# Manipulate-Anything: Automating Real-World Robots using Vision-Language Models

**Jiafei Duan** [1*] **Wentao Yuan** [1*] **Wilbert Pumacay** [2] **Yi Ru Wang** [1]
**Kiana Ehsani** [3] **Dieter Fox** [1,4] **Ranjay Krishna** [1,3]
[1]University of Washington [2]Universidad Católica San Pablo
[3]Allen Institute for Artificial Intelligence [4]NVIDIA

**Abstract:** Large-scale endeavors like RT-1[1] and widespread community efforts such as Open-X-Embodiment [2] have contributed to growing the scale of robot demonstration data. However, there is still an opportunity to improve the quality, quantity, and diversity of robot demonstration data. Although vision-language models have been shown to automatically generate demonstration data, their utility has been limited to environments with privileged state information, they require hand-designed skills, and are limited to interactions with few object instances. We propose MANIPULATE-ANYTHING[2], a scalable automated generation method for real-world robotic manipulation. Unlike prior work, our method can operate in real-world environments without any privileged state information, hand-designed skills, and can manipulate any static object. We evaluate our method using two setups. First, MANIPULATE-ANYTHING successfully generates trajectories for all 7 real-world tasks and 14 simulation tasks, significantly outperforming state-of-the-art methods such as VoxPoser. Second, MANIPULATE-ANYTHING's demonstrations can train more robust behavior cloning policies than training with human demonstrations, or from data generated by VoxPoser [3], Scaling-up [4] and Code-As-Policies [5]. We believe MANIPULATE-ANYTHING can be the scalable method for both generating data for robotics and solving novel tasks in a zero-shot setting. Project page: robot-ma.github.io.

**Keywords:** Zero-shot data generation, vision-language models, robot skill generation, behavior cloning, and robotic manipulation

## 1   Introduction

The success of modern machine learning systems fundamentally relies on the *quantity* [6, 7, 8, 9, 10, 11], *quality* [12, 13, 14, 15, 16], and *diversity* [17, 18, 19, 20, 21] of the data they are trained on. The availability of large-scale internet data made possible significant advances in vision and language [22, 23, 24]. However, the dearth of data has prevented similar advancements in robotics. Human demonstration collection methods do not scale to sufficient *quantity* or *diversity*. Projects like RT-1 [1] demonstrated the utility of high-*quality* human data collected over 17 months. Others have developed low-cost hardware for data collection [25, 26, 27]. However, all these procedures require expensive human data collection.

Automated data collection methods do not scale to sufficient *diversity*. With the advent of vision-language models (VLMs), the robotics community has been abuzz with new systems that leverage VLMs to guide robotic behavior [28, 5, 29, 30, 3, 4, 31]. In these systems, VLMs decompose tasks into language plans [28, 5] or generate code to execute predefined skills [32, 3]. Though successful in simulation, these methods underperform in the real world [32, 3]. Some methods rely on privileged

---

[*]Equal contribution
[2]Mainly suited for solving most quasistaic, prehensile, and some non-prehensile tasks.

8th Conference on Robot Learning (CoRL 2024), Munich, Germany.

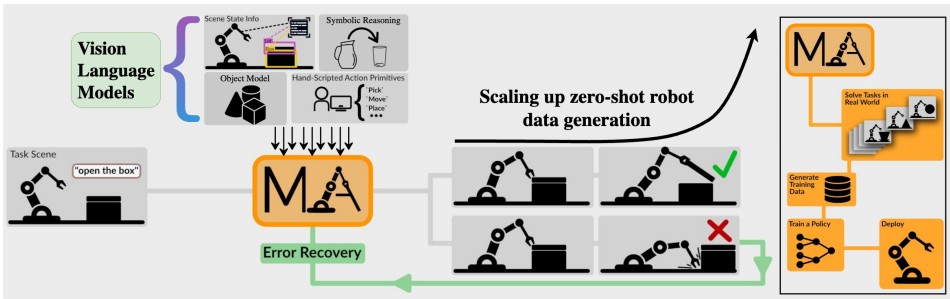

Figure 1: MANIPULATE-ANYTHING is an automated method for robot manipulation in real world environments. Unlike prior methods, it does not require privileged state information, hand-designed skills, or limited to manipulating a fixed number of object instances. It can guide a robot to accomplish a diverse set of unseen tasks, manipulating diverse objects. Furthermore, the generated data enables training behavior cloning policies that outperform training with human demonstrations.

state information only available in simulation [29, 4], require hand-designed skills [30], or are also limited to manipulating a fixed set of object instances with known geometric shape [3, 32].

As VLMs improve in performance, and with the vast common-sense knowledge they have shown to possess, could we harvest their capabilities for diverse task completion and scalable data generation? The answer is yes – with careful system design and the right set of input and output formulations, we can not only use VLMs as a means to successfully perform *diverse* tasks in a zero-shot manner, but also generate *quality* data at a high *quantity* to train behaviour cloning policies.

**We propose MANIPULATE-ANYTHING a scalable automated demonstration generation method for real-world robotic manipulation.** MANIPULATE-ANYTHING produces high *quality* data, at large-*quantities* (if needed), and can manipulate a *diverse* set of objects to perform a *diverse* set of tasks. When placed in a real world environment and given a task (e.g., "open the top drawer" in Figure 2), MANIPULATE-ANYTHING effectively leverages VLMs to guide a robotic arm to complete the task. Unlike prior methods, it doesn't need privileged state information, hand-designed skills, or limited to specific object instances. Not relying on privileged information makes MANIPULATE-ANYTHING environment-agnostic. MANIPULATE-ANYTHING plans a sequence of sub-goals and generates actions to execute the sub-goals. It can verify whether the robot succeeded in the sub-goal using a verifier and re-plan from the current state if needed. This error recovery enables mistake identification, re-planning, and recovering from failure. It also injects recovery behavior into the collected demonstrations. We further enhanced the VLMs' capabilities by incorporating reasoning from multi-viewpoints, significantly improving performance.

We showcase the utility of MANIPULATE-ANYTHING through two evaluation setups. First, we show that it can be prompted with a novel, never-before-seen task and complete it in a zero-shot manner. We quantitatively evaluate across 7 real-world and 14 RLBench [33] simulation tasks and demonstrate capabilities across many real-world everyday tasks (refer to appendix). Our method significantly outperforms VoxPoser [3] in 10/14 simulation tasks for zero-shot evaluation. It also generalizes to tasks where VoxPoser completely fails because of its limitation to specific object instances. Furthermore, we demonstrated that our approach can solve real-world manipulation tasks in a zero-shot manner, achieving a task-averaged success rate of 38.57%. Second, we show that MANIPULATE-ANYTHING can generate useful training data for a behavior cloning policy. We compare MANIPULATE-ANYTHING generated data against ground truth hand-scripted demonstrations as well as against data from VoxPoser [3], Scaling-up [4] and Code-As-Policies [5]. Surprisingly, policies trained on our data outperforms even human hand-scripted data on 4 out of 12 tasks and performs on par for 3 more when trained with PerAct [34]. Meanwhile, the baselines are unable to generate the training data for some of tasks. MANIPULATE-ANYTHING demonstrates the broad possibility of large-scale deployment of robots across unstructured real-world environments. It also highlights its utility as a training data generator, aiding in the crucial goal of scaling up robot demonstration data.

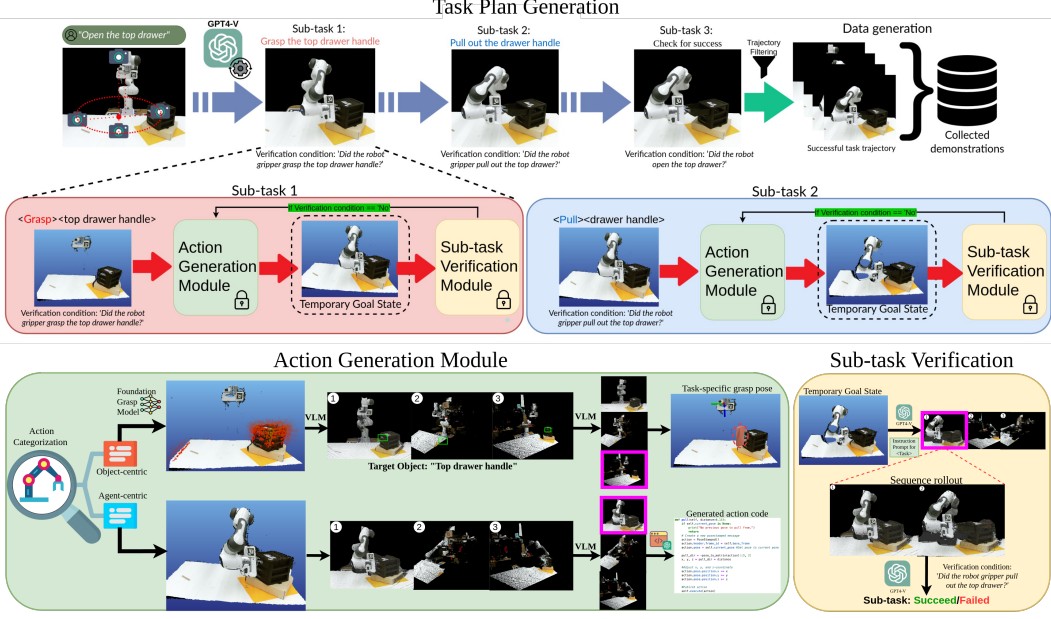

Figure 2: **Manipulate Anything Framework.** The process begins by inputting a scene representation and a natural language task instruction into a VLM, which identifies objects and determines sub-tasks. For each sub-task, we provide multi-view images, verification conditions, and task goals to the action generation module, producing a task-specific grasp pose or action code. This leads to a temporary goal state, assessed by the sub-task verification module for error recovery. Once all sub-tasks are achieved, we filter the trajectories to obtain successful demonstrations for downstream policy training.

## 2 Related work

MANIPULATE-ANYTHING enables scaling of robotic manipulation data using VLMs. As such, we review recent efforts in 1) scaling manipulation data, and 2) applications of VLMs in robotics.

**Scaling manipulation data.** When deploying vision and language-based control policies for real-world applications, a significant challenge revolves around acquiring data. Traditionally, a convenient avenue to collect such trajectories is through human annotations for action (i.e. through teleoperation) and language labeling [34, 35, 36], however, this approach is limited in scale. To address this limitation and achieve autonomous scalability, prior works employ vision-language models or procedurally generate language annotations in simulated environments [4, 37, 38]. For action labels, strategies range from random exploration to learned policies [39]. While human egocentric videos are relevant, they lack action labels and require cross-embodiment transfer [40]. Another strategy involves model-based policies, such as task and motion planning (TAMP) [41]. Our approach extends these methods by incorporating common-sense knowledge from LLMs and VLMs, providing a framework that combines the strengths of VLMs, object pose prediction, and dynamic retry to synthesize demonstrations in simulated and real environments.

**Language models for robotics.** In the field of robotics, large language models have found diverse applications, including policy learning [42, 43, 44], task and motion planning [45, 46], log summarization [47], policy program synthesis [5], and optimization program generation [28]. Previous research has also explored the physical grounding capabilities of these models [3, 32, 48, 49], while ongoing work [4] investigates their integration with task and motion planners to create expert demonstrations. [36] attempted to collect extensive real-world interaction data, with short-horizon trajectories. [50] proposed a key-point based visual prompting method for real-world manipulation, through predicting affordances and corresponding motions. Our work complements the existing line of works by leveraging the high-level planning capabilities of language models, scene understanding capabilities of vision language models, and action sampling to enable synthesis of robot trajectories, which include language, vision, and robot state, given arbitrary tasks and environments.

# 3 MANIPULATE-ANYTHING

We propose MANIPULATE-ANYTHING, a framework that solves everyday manipulation tasks conditioned on language. Under the hood, MANIPULATE-ANYTHING leverages VLMs to decompose tasks into sub-tasks, generates code for new skills or task-specific grasp pose, and verifies the success of each sub-task (Figure 3). Note that due to the modularity aspect of our framework, MANIPULATE-ANYTHING will continue to improve as the underlying VLMs continue to improve.

## 3.1 Task plan generation

MANIPULATE-ANYTHING takes as input any task described by a free-form language instruction, $\mathbf{T}$ (e.g., *'open the top drawer'*). Creating robot trajectories that adheres to $\mathbf{T}$ is challenging due to its potential complexity and ambiguity, requiring a nuanced understanding of the current environment state. Given $\mathbf{T}$, and an image of the scene, we apply a VLM to first identify task-relevant objects in the scene, appending them to a list. Subsequently, we use a LLM along with those information to decompose the main task $\mathbf{T}$ into a series of discrete, smaller sub-tasks, represented as $\mathbf{T}_i$, along with the corresponding verification conditions $v_i$, where $i$ ranges from 1 to $n$. For instance, the above task could be decompose into sub-tasks include *'grasp the drawer handle'* or *'pull open the drawer'*, and verification conditions are *'did the robot grasp the handle?'* or *'is the drawer opened?'*. This transforms the instruction $\mathbf{T}$ into a sequence of specific sub-tasks $\{(\mathbf{T}_1, v_1), (\mathbf{T}_2, v_2), \ldots, (\mathbf{T}_n, v_n)\}$. For each sub-task, MANIPULATE-ANYTHING generates desired actions (§ 3.3) and verifies them against the corresponding conditions to ensure successful completion of that sub-task(§ 3.4). This verification step also allows MANIPULATE-ANYTHING to recover from mistakes and attempt again in the case of failure.

## 3.2 Multi-viewpoint VLM selection

Many prior works that investigated VLM's reasoning capability found that they do not work well given a single viewpoint [51, 52, 53]. In robotic manipulation, we leverage multiple viewpoints from either more than one camera or re-rendering to minimize object occlusion [54, 55]. Leveraging these insights, we proposed a multi-viewpoint selection phase via VLM before using the selected viewpoints for either action generation or sub-task verification for MANIPULATE-ANYTHINGWe concatenate all available viewpoints from the current observations, either from multiple cameras or re-rendered viewpoints from a single RGB-D, into a single frame. Numbers are annotated on the top left of each concatenated frame to correspond to a specific viewpoint. Using the concatenated multi-viewpoints, we then query VLMs to choose an ideal viewpoint conditioned on the sub-task. Therefore, for both agent-centric and object-centric action generation, we render multiple viewpoints of the scene and query VLMs to choose an ideal viewpoint for either generating actions given the sub-task or verifying if the sub-task verification condition has been met as shown in Figure 3.1

## 3.3 Action generation module

Given a sub-task, the desired output from the action generation module is a sequence of low-level actions represented as a 6 DoF end-effector pose. The actions can be categorized into two sets: agent-centric or object-centric. Based on the generated sub-task, the LLM planner will classify it as either agent-centric or object-centric actions. For agent-centric actions, it will modify the agent's state; e.g., it can move the robot's end-effector from the current state (e.g., "rotate $90°$"). We first employed our multi-viewpoint selection to sample out the most optimal viewpoints to provide the VLM with along with the in-context learning technique to generate the code for synthesizing the desired motion. Unlike prior methods that use language models solely for code generation [5], we leverage VLMs to reason about object locations and the scene, grounding generation in the current state. This advantage is demonstrated in the ablation studies in the Appendix.

For object-centric actions, it is often used to generate a task-specific grasp pose for grasping a certain object (e.g., "grasp a knife for cutting") or to synthesize a pre-action pose for non-prehensile tasks by allowing the VLM to assign translation offsets to task-specific grasp poses. To obtain an object-centric action for a given sub-task, we first use an **object-agnostic grasp prediction model** [56] to generate all possible 6-DOF grasping poses in the scene. These poses are not conditioned on task specifications and may include invalid grasp poses for the given task. Next, we use the VLM

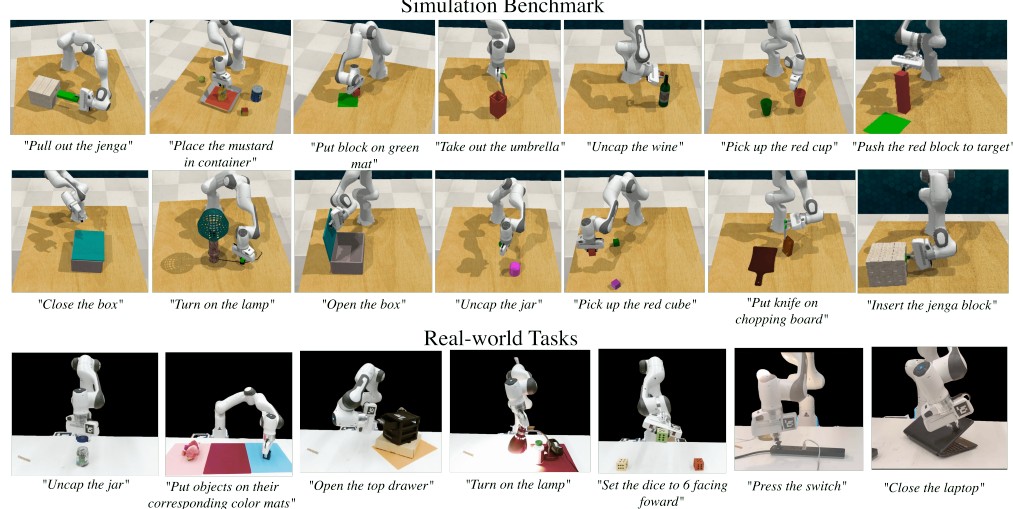

Figure 3: MANIPULATE-ANYTHING is an open-vocabulary robot demonstration generation system. We show zero-shot demonstrations for 14 tasks in simulation and 7 tasks in the real world.

to obtain the bounding box of the target object parts conditioned on the task specification from all available or re-rendered viewpoints (e.g., if the task is "grasp a knife," the VLM will detect the handle of the knife and generate a bounding box for the handle). The VLM then performs multi-viewpoint selection to identify the most optimal viewpoint free from occlusion. Finally, using the bounding box detection of the task-specific part of the object from the optimal viewpoint selected by the VLM, we filter out a list of proposed candidate grasp poses and select the highest confidence grasp pose. This approach allows us to obtain the most optimal task-specific grasp pose, placement pose, and pre-action pose for non-prehensile tasks, all leveraging the capabilities of the VLM. After the action is generated, a motion planner can be used to move the robot to the desired pose as detailed in Fig. 2.

### 3.4 Sub-task verification

To ensure that each sub-task $T_i$ is executed correctly, we introduce a VLM-based verifier. After every action for each sub-task is executed, we use the VLM to check if the end state matches the verifier condition $v_i$. Similar to the action generation module, we use **multi-viewpoint VLM selection** to find the optimal view, avoiding errors due to occlusion or ambiguity from a single viewpoint. If the verifier identifies failure, we re-attempt the action generation step for the previous sub-task from the current state. Otherwise, the next sub-task $T_{i+1}$ is attempted. More details are in the appendix 6.1.

## 4 Experiments

Our experiments are designed to address two questions: 1) Can MANIPULATE-ANYTHING accurately solve a diverse set of tasks in a zero-shot manner? 2) Can data generated from MANIPULATE-ANYTHING be used to train a robust policy?

**Implementation details.** We use both GPT-4V and Qwen-VL [57] as our VLM. We use GPT-4V for task decomposition, action generation, and verification. We use Qwen-VL to detect and extract object information. To ensure zero-shot execution within a reasonable budget, we limit the number of action steps in each trajectory to 50 and the verification module allows a maximum of 30 tries to accomplish a sub-goal. For the task plan generation, we follow the prompting structure adapted from ProgPrompt [28]. All prompts input into the VLM are accompanied by few-shot demonstrations [58]. Additionally, we provide three manually curated primitive action code snippets as examples to prompt the VLM for new action code generation. Full prompts are included in the Appendix. We use four viewpoints $\mathbf{M}_4 = [front, wrist, left\_shoulder, right\_shoulder]$ for the simulation experiments, and re-render three viewpoints for the real-world experiments [54]. For better reasoning by the VLM, we use a resolution of $256 \times 256$.

### 4.1 Zero-shot Performance in Simulation

We empirically study the zero-shot capability of MANIPULATE-ANYTHING in solving 14 diverse tasks in simulation, covering a wide range of task configurations and action primitives for both prehensile

Table 1: **Task-averaged success rate % for zero-shot evaluation.** MANIPULATE-ANYTHING outperformed other baselines in 10 out of 14 simulation tasks from RLBench [33]. Each task was evaluated over 3 seeds to obtain the task-averaged success rate and standard deviations.

| Method | Put_block | Play_jenga | Open_jar | Close_box | Open_box | Pickup_cup | Push_block |
|---|---|---|---|---|---|---|---|
| VoxPoser [3] | 70.70±2.31 | 0.00±0.00 | 0.00±0.00 | 0.00±0.00 | 0.00±0.00 | 26.70±14.00 | **25.33**±8.33 |
| CAP [5] | 84.00±16.00 | 0.00±0.00 | 0.00±0.00 | 0.00±0.00 | 0.00±0.00 | 14.67±4.62 | 8.00±4.00 |
| Scaling-up [4] | 77.33±6.11 | 0.00±0.00 | 78.67±11.55 | 0.00±0.00 | 0.00±0.00 | 9.33±2.26 | 5.33±6.11 |
| MA (Ours) | **96.00**±4.00 | **77.33**±6.11 | **80.00**±4.00 | 33.33±12.86 | 29.00±10.07 | **82.67**±14.04 | 20.00±4.00 |

| Method | Take_umbrella | Sort_mustard | Open_wine | Lamp_on | Put_knife | Pick_&_lift | Insert_block |
|---|---|---|---|---|---|---|---|
| VoxPoser[3] | 33.33±8.33 | **96.00**±6.93 | 8.00±4.00 | 57.30±12.22 | **92.00**±4.00 | 96.00±0.00 | 0.00±0.00 |
| CAP[5] | 4.00±4.00 | 0.00±0.00 | 0.00±0.00 | 64.00±6.93 | 14.67±8.33 | **100.00**±0.00 | 0.00±0.00 |
| Scaling-up [4] | 6.67±2.31 | 41.33±12.86 | 33.33±20.13 | 60.00±8.00 | 24.00±0.00 | **100.00**±0.00 | 0.00±0.00 |
| MA (Ours) | **61.33**±20.13 | 64.00±6.93 | **42.00**±4.00 | **69.33**±6.11 | 52.00±10.58 | 84.00±6.93 | **33.33**±4.62 |

and non-prehensile tasks. Our simulation experiments are reported to ensure reproducibility and provide a benchmark for future methods.

**Environment and tasks.** The simulation setup involves a Franka Panda robot with a parallel gripper, using CoppeliaSim and PyRep for interfacing. Four RGB-D cameras capture input observations around a tabletop environment. RLBench [33] is used as the task benchmark, with 14 sampled tasks that cover various action primitives, task horizons, and object position variations. The robot's actions are represented as waypoints, with trajectories computed and executed via a motion planner [59].

**Baselines.** We compare against three state-of-the-art zero-shot data generation approaches: Code-as-Policies (CAP) [5], Scaling-up-Distilling-Down (Scaling-up) [4] and VoxPoser [3]. CAP uses language models to generate programs that call hand-crafted primitive actions, while VoxPoser predicts waypoints via a 3D voxel map of value functions. Scaling-up leverages an LLM with 6 DoF exploration primitives to generate robotic data for policy distillation. We provided CAP and Scaling-up with ground truth simulation states and object models, and VoxPoser with segmented object point clouds, which inherently disadvantages MANIPULATE-ANYTHING in comparison.

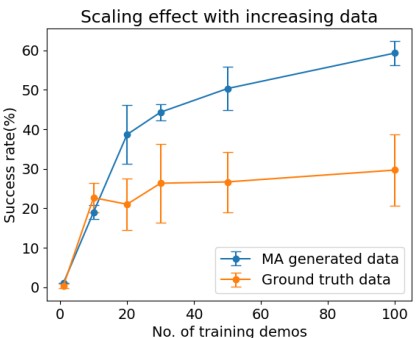

Figure 4: **Scaling experiment.** Scaling effect of model performance with increasing training demonstrations.

**Results:** **MANIPULATE-ANYTHING can generate successful trajectories for all** 14 **tasks while Scaling-up, VoxPoser and CAP cover only** 10, 9 **and** 7 tasks, respectively (Table 1). MANIPULATE-ANYTHING outperforms the baselines in 10 out of the 14 tasks. The three lowest-performing tasks by MANIPULATE-ANYTHING are non-prehensile or complex fine-grained manipulation tasks that require generating new primitive actions with precise parameters. VoxPoser fails in the tasks that require moving the arm beyond 4-DoF. MANIPULATE-ANYTHING outperforms the strongest baseline, VoxPoser, by an average task-averaged margin of up to 22%.

## 4.2 Behavior cloning with demonstrations from MANIPULATE-ANYTHING

Next, we analyze the quality of the generated data by comparing the success rates of behavior cloning models trained with the data. Zero-shot methods like MANIPULATE-ANYTHING are computationally expensive but hold the potential to generate useful training data. To evaluate the quality and effectiveness of the generated training data, we use the methods described in the previous section to generate data for each task. We also compare performance against a model trained on human-generated demonstrations across the 12 tasks. We use the data to train behavior cloning policies.

Table 2: **Behavior Cloning with different generated data.** The behavior cloning policy trained on the data generated by MANIPULATE-ANYTHING provides the best performance on 10 out of 12 tasks compared to the other autonomous data generation baselines (excluding RLBench). We report the Success Rate % for behaviour cloning policies trained with data generated from VoxPoser [3] and Code as Policies [5] in comparison. Note that the RLBench[33] baseline uses human expert demonstrations and is considered an upper bound for behavior cloning.

| Data | Models | Put_block | Play_jenga | Open_jar | Close_box | Open_box | Pickup_cup |
|---|---|---|---|---|---|---|---|
| VoxPoser[3] | PerAct[34] | 2.67±2.31 | - | - | - | - | 4.00±4.00 |
| CAP[5] | PerAct[34] | 6.67±2.31 | - | - | - | - | 14.67±12.86 |
| Scaling-up [4] | PerAct[34] | 22.67±15.14 | - | 5.33±6.11 | - | - | 14.67±2.31 |
| MA (Ours) | PerAct[34] | **85.33**±10.07 | 81.33±2.31 | 21.33±10.07 | 42.67±8.33 | **30.67**±11.55 | 54.00±12.49 |
| RLBench[33] | PerAct[34] | 20.00±18.33 | **81.33**±9.24 | **58.67**±45.49 | **68.00**±24.98 | 14.67±6.11 | **54.67**±23.09 |

| Data | Models | Take_umbrella | Sort_mustard | Open_wine | Lamp_on | Put_knife | Pick_&_lift |
|---|---|---|---|---|---|---|---|
| VoxPoser[3] | PerAct[34] | 4.00±4.00 | 0.00±0.00 | 1.33±2.31 | 5.33±4.62 | 1.33±2.31 | 5.67±1.64 |
| CAP[5] | PerAct[34] | 13.33±10.06 | - | - | 8.00±16.00 | 9.33±6.11 | 46.67±2.31 |
| Scaling-up [4] | PerAct[34] | 4.00±4.00 | 0.00±0.00 | 81.33±12.86 | 76.00±4.00 | 5.33±2.31 | 53.33±10.06 |
| MA (Ours) | PerAct[34] | **84.00**±6.93 | 53.33±6.11 | 86.67±6.11 | **89.33**±6.11 | 8.00±4.00 | 33.33±2.31 |
| RLBench[33] | PerAct[34] | 58.67±50.80 | **53.33**±34.02 | **86.67**±12.86 | 84.00±13.86 | **30.67**±10.07 | **62.67**±9.24 |

**Data generation details.** We generate 10 successful demonstrations per task. We use the system's success condition to filter for successful demonstrations. Each of the demonstrations consist of a language instruction, RGB-D frames for the trajectory, and waypoints represented as 6 DoF gripper poses and states. For the tasks that the baselines were unable to generate any successful demonstrations, we patched the missing training data with RLBench system-generated demonstrations.

**Training and evaluation protocol.** We train two models using the generated demonstrations: the Perceiver-Actor (PerAct) model [34], a transformer-based robotic manipulation behavior cloning model that expects tokenized voxel grids and language instructions as inputs and predicts discretized voxel grid 6 DoF poses and gripper states, and RVT-2 model [60], a multi-view transformer-based BC model. The RVT-2 model uses tokenized image patches and CLIP-encoded language instructions as input to predict keypoint actions as translation heatmaps, discretized rotation in Euler angles, and the gripper's binary state. Notably, RVT-2 is currently the highest-performing model on the RLBench benchmark. For all the generated training datasets, we train a multi-task PerAct policy with a batch size of 4 for 30k iterations on a single RTX A100, and RVT-2 with a batch size of 24 for 3.3k iterations on 4 A100s. To ensure consistent evaluation, we generate one set of testing environments with RLBench. We evaluate the last checkpoint from each of the trained policies. Each policy is evaluated for 25 episodes across each task using 3 different seeds. We measure the success rate based on the simulation-defined success condition. Details on RVT-2 results can be found in the Appendix.

**Results: Policies trained using MANIPULATE-ANYTHING data perform similarly to policies trained using hand-scripted demonstrations ($p = 0.973$) for PerAct (Table 2).** Training on either MANIPULATE-ANYTHING or hand-scripted demonstrations results in a performance difference of just 0.27% average across all tasks. Furthermore, models trained on data from the baselines exhibit a statistically lower performance ($p \leq 0.01$ for VoxPoser, Scaling-up and CAP). One of the main factors potentially contributing to the performance differences could be that MANIPULATE-ANYTHING generates diverse expert trajectories preferable to humans. This is shown in Fig. 11, which illustrates the action distribution of generated data by different methods for the same given tasks. We also observed that the policy trained on MA data achieves a lower standard deviation of 7.19 average across all tasks, compared to the zero-shot performance standard deviation of 8.81. This suggests the benefits of training over generated data instead of relying solely on zero-shot deployment.

### 4.3 Real-world experiments

**Environment and tasks.** We employ a Franka Panda manipulator equipped with a parallel gripper. We use a front-facing Kinect 2 RGB-D camera. To generate multi-view inputs for the

Table 3: **Real-world Results.** The model trained on the data generated by our model in the real world (no expert in the loop) demonstrates on par results with the model trained on human expert collected data. We present a comparison of success rates for task completion in a zero-shot manner (Code as Policies [5] and MANIPULATE-ANYTHING), and using trained policies from MANIPULATE-ANYTHING data and human expert data.

| | Open_drawer | Sort_object | On_lamp | Open_jar | Correct_dice | Press_switch | Close_laptop |
|---|---|---|---|---|---|---|---|
| CAP (0-shot) | $0.00 \pm 0.00$ | $13.33 \pm 5.77$ | $0.00 \pm 0.00$ | $6.67 \pm 5.77$ | $6.67 \pm 5.77$ | $0.00 \pm 0.00$ | $0.00 \pm 0.00$ |
| MA (0-shot) | $\textbf{36.67}\pm5.77$ | $\textbf{60.00}\pm10.00$ | $\textbf{26.67}\pm11.55$ | $\textbf{40.00}\pm10.00$ | $\textbf{53.33}\pm5.77$ | $\textbf{20.00}\pm10.00$ | $\textbf{33.33}\pm5.77$ |
| PerAct (MA data) | $50.00 \pm 0.00$ | $33.33 \pm 5.77$ | $50.00 \pm 0.00$ | $56.67 \pm 5.77$ | $60.00 \pm 0.00$ | $56.67 \pm 5.77$ | $33.33 \pm 5.77$ |
| PerAct (Human data) | $\textbf{53.33}\pm11.55$ | $\textbf{36.67}\pm5.77$ | $\textbf{60.00}\pm0.00$ | $\textbf{76.67}\pm5.77$ | $\textbf{80.00}\pm10.00$ | $33.33\pm5.77$ | $\textbf{53.33}\pm5.77$ |

MANIPULATE-ANYTHING framework, we re-render virtual viewpoints from the generated point cloud, similar to prior work [54]. We selected 7 representative real-world tasks, both prehensile and non-prehensile: `open_jar`, `sort_objects`, `correct_dices`, `open_drawer`, `on_lamp`, `press_switch`, and `close_laptop`, all conditioned on language instructions. Each task was evaluated over 10 episodes with varying object poses across 3 trials.

**Data generation details.** We used MANIPULATE-ANYTHING to generate 6 demonstrations for each task and manually perform scene resets when failures occur. We train a similar multi-task PerAct for 120k iterations and evaluate the trained policies in a manner similar to the zero-shot experiments.

**Results: MANIPULATE-ANYTHING is able to generate successful demonstrations for each of the 7 real world tasks.** Even for the worst-performing task, MANIPULATE-ANYTHING achieves a success rate of more than 25%. Our approach outperforms **CAP** by 38%. Consistent with the simulation results, **training with the data generated by MANIPULATE-ANYTHING produces a more robust policy** compared to performing zero-shot. Additionally, in 4 out of 5 tasks, the trained policies perform better than the zero-shot approach. The policy under-performs on the `sort_object` task, because it requires longer-horizon memory —a known limitation pointed out in PerAct [34].

## 4.4 Ablations

For effective real-world deployment of MANIPULATE-ANYTHING, it's crucial that the collected data supports scaling of robotics transformers and offers diverse skills and interacted objects. We conducted an ablation study to evaluate the quality of MANIPULATE-ANYTHING-generated data for scaling and its generalization to language instruction changes. For scaling, we generated behavior cloning data, ranging from 1 to 100 training demonstrations from RLBench and MANIPULATE-ANYTHING for a single task (`put_block`), and trained a PerAct policy. For generalization, we varied the `sort_mustard` task with different language instructions and target objects. We compared our approach to VoxPoser to assess robustness to object and language instruction changes. Further implementation details are in the supplementary materials. **Result:** Our scaling experiments demonstrate that generating more training data via MANIPULATE-ANYTHING improves PerAct policy performance (Fig. 4). The data from our approach shows a better rate of change with a slope of 0.503 for a linear fit, compared to 0.197 for RLBench-generated data. Additionally, MANIPULATE-ANYTHING data is more generalizable and robust to language instruction changes, outperforming VoxPoser in task success across language and object variations. Detailed results in the appendix.

## 5 Discussion

**Limitations.** Despite promising results, MANIPULATE-ANYTHING has limitations: reliance on large language models (VLMS), which may lessen with the development of open-source VLMS; inability to generate alternative grasp poses for dynamic manipulation and non-prehensile tasks; potential compounding errors from integrating multiple VLMs, possibly mitigated by emerging specialized VLMs; and the need for manual prompt engineering for in-context learning, with advancements in alignment and prompting techniques potentially reducing this effort. **Conclusion.** MANIPULATE-ANYTHING is a scalable, environment-agnostic approach that leverages VLMs for high-level planning, scene understanding, and error recovery to generate zero-shot demonstrations for robotic tasks without privileged environment information. This results in high-quality data for behavior cloning, outperforming human data.

**Acknowledgments**

Jiafei Duan is supported by the Agency for Science, Technology and Research (A*STAR) National Science Fellowship. Wilbert Pumacay is supported by grant 234-2015-FONDECYT from Cienciactiva of the National Council for Science, Technology and Technological Innovation(CONCYTEC-PERU). We would also like to thank Winson Han for the help in designing of the figures. This project was partially funded by an Amazon Science grant. This project is also sponsored by RACER2 Project: HR001123C0150.

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

# 6 Appendix

## 6.1 MANIPULATE-ANYTHING implementations

**Action Generation Module.** We generate each action using either an agent-centric or object-centric approach. For object-centric action generation, we utilize M2T2 [56], NVIDIA's foundational grasp prediction model, for `pick` and `place` actions. For 6-DoF grasping, we input a single 3D point cloud from either a single RGB-D camera (in the real world) or multiple cameras (in simulation). The model outputs a set of grasp proposals on any graspable objects, providing 6-DoF grasp candidates (3-DoF rotation and 3-DoF translation) and default gripper close states. For placement actions, M2T2 outputs a set of 6-DoF placement poses, indicating where the end-effector should be before executing a `drop` primitive action based on a VLM plan. The network ensures the object is stably positioned without collisions. We also set default values for `mask_threshold` and `object_threshold` to control the number of proposed grasp candidates. After proposing a list of template grasp poses, we use QWen-VL [57] to detect the target object by prompting the current image frame with the target object's name, translated into Chinese using a machine translation model [61]. This detection is applied to all re-rendered viewpoints or viewpoints from different cameras. We then concatenate these frames into a single image, annotating each sub-image with a number at the top right corner. Next, we call the GPT-4V API with few-shot demonstrations and the task goal to prompt GPT-4V to output the selected number of viewpoints that provide the most unobstructed views for sampling the grasp pose to achieve the sub-goal. Using the selected viewpoint, we execute the grasp by moving the end-effector to the sampled grasp pose via a motion planner.

For agent-centric action generation, we first perform the same steps of viewpoint selection. Using the selected viewpoint, a few demonstration examples, and the sub-goal, we prompt GPT-4V to generate an action function with code snippets that include the necessary code to perform a delta-action on the current robot pose. We then execute this by moving the end-effector based on these delta changes. This process is iterated until we obtain the most desirable code snippet function for the given sub-goals, which is then appended to a skill library for future use.

**Sub-goal Verification Module** The sub-goal verification module helps with error recovery by ensuring all potential attempts at resolving the current step action have been tried. With the temporary goal state obtained by the action generation module, we use multi-viewpoints to sample the optimal viewpoint for answering the verification condition generated for the given sub-goal during the task plan generation phase. Using the same viewpoint selection method as in the Action Generation Module, we obtain the optimal view and then perform a two-step sequence rollout of frames: one from the current frame at this viewpoint and another from the previous action step. We concatenate these two frames, annotate them with numbers to indicate their temporal relation, and use this image to prompt GPT-4V to check if the verification condition is fulfilled as shown in Fig. 5. If the answer is `Yes`, we proceed to the next sub-goal. If the answer is `No`, we resample new viewpoints, generate new actions, and reattempt the entire sub-goal with a different seed.

## 6.2 Simulation experiments

**Simulation Setup.** All the simulated experiments use a four-camera setup as illustrated in Fig. 7. The cameras are positioned at the front, left shoulder, wrist, and right shoulder. All cameras are static,

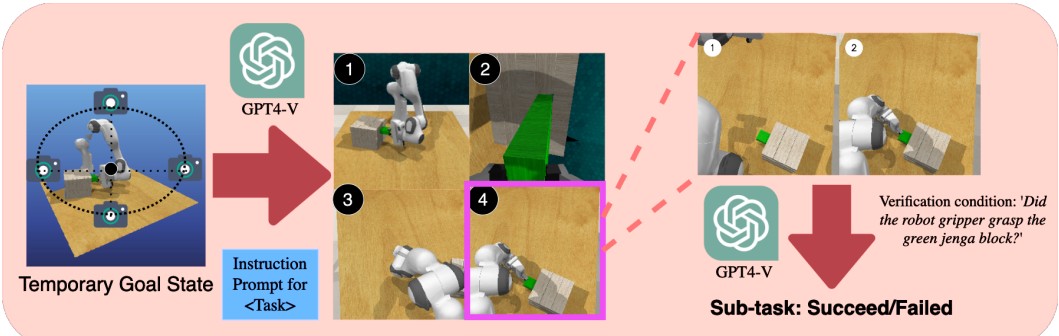

Figure 5: **Sub-goal Verification Module.** We used viewpoint selection similar to action generation, to find the optimal viewpoints, and roll out the two step sequences of the previous and current frames for prompting the verification condition.

except for the wrist camera, which is mounted on the end effector. We did not modify the default camera poses from the original RLBench [33]. These poses maximize coverage of the entire table, and we use a 256 x 256 resolution for better input to the VLMs.

**Task Details.** We describe in detail each of the 12 tasks for simulation evaluation, both for trained policies and zero-shot methods, along with their RLBench variations and success conditions. We have made some modifications to the original tasks to enhance the detection rate by Code-As-Policies and VoxPoser.

**Additional Simulation Results.** In addition to evaluating the generated data using PerAct [34], we also assessed it with RVT-2 [60], the current state-of-the-art model on the RLBench benchmark. While the results of models trained with RVT-2 were significantly higher than those trained with PerAct, the overall trends and performance patterns remained consistent across the same training data as depicted in Table 4.

### 6.2.1 `put block`

**Filename:** `put_block.py`

**Task:** Pick up the green block and place it on the red mat.

**Success Metric**: The success condition on the red mat detects the target green block.

### 6.2.2 `close box`

**Filename:** `close_box.py`

**Task:** Close the box.

**Success Metric**: The revolute joint of the specified handle is at least $60°$ off from the starting position.

### 6.2.3 `open box`

**Filename:** `open_box.py`

**Task:** Open the box.

**Success Metric**: The revolute joint of the specified handle is at least $60°$ off from the starting position.

### 6.2.4 `play jenga`

**Filename:** `play_jenga.py`

**Task:** Pull out the green jenga block.

**Success Metric**: The green jenga block is out of its pre-defined location.

Table 4: **Behavior Cloning with different generated data.** The behavior cloning policy trained on the data generated by MANIPULATE-ANYTHING provides the best performance on 10 out of 12 tasks compared to the other autonomous data generation baselines. We report the Success Rate % for behaviour cloning policies trained with data generated from VoxPoser [3] and Code as Policies [5] in comparison. Note that the RLBench [33] baseline uses human expert demonstrations and is considered an upper bound for behavior cloning.

| Data | Models | Put_block | Play_jenga | Open_jar | Close_box | Open_box | Pickup_cup |
|------|--------|-----------|------------|----------|-----------|----------|------------|
| VoxPoser [3] | PerAct [34] | 2.67±2.31 | - | - | - | - | 4.00±4.00 |
| CAP [5] | PerAct [34] | 6.67±2.31 | - | - | - | - | 14.67±12.86 |
| Scaling-up [4] | PerAct [34] | 22.67±15.14 | - | 5.33±6.11 | - | - | 14.67±2.31 |
| MA (Ours) | PerAct [34] | **85.33**±10.07 | 81.33±2.31 | 21.33±10.07 | 42.67±8.33 | **30.67**±11.55 | 54.00±12.49 |
| RLBench [33] | PerAct [34] | 20.00±18.33 | **81.33**±9.24 | **58.67**±45.49 | **68.00**±24.98 | 14.67±6.11 | **54.67**±23.09 |
| VoxPoser [3] | RVT-2 [60] | 73.33±2.31 | - | - | - | - | 2.67±2.31 |
| CAP [5] | RVT-2 [60] | 78.66±8.32 | - | - | - | - | 77.33±19.73 |
| Scaling-up [4] | RVT-2 [60] | 38.67±2.31 | - | 33.33±2.31 | - | - | 92.00±4.00 |
| MA (Ours) | RVT-2 [60] | 85.33±2.31 | 82.67±2.31 | 78.67±10.06 | **82.67**±2.31 | 24.00±4.00 | 97.33±2.31 |
| RLBench [33] | RVT-2 [60] | **86.67**±2.31 | **85.33**±2.31 | **81.33**±6.11 | 76.00±4.00 | 4.00±4.00 | **97.33**±2.31 |

| Data | Models | Take_umbrella | Sort_mustard | Open_wine | Lamp_on | Put_knife | Pick_&_lift |
|------|--------|---------------|--------------|-----------|---------|-----------|-------------|
| VoxPoser [3] | PerAct [34] | 4.00±4.00 | 0.00±0.00 | 1.33±2.31 | 5.33±4.62 | 1.33±2.31 | 5.67±1.64 |
| CAP [5] | PerAct [34] | 13.33±10.06 | - | - | 8.00±16.00 | 9.33±6.11 | 46.67±2.31 |
| Scaling-up [4] | PerAct [34] | 4.00±4.00 | 0.00±0.00 | 81.33±12.86 | 76.00±4.00 | 5.33±2.31 | 53.33±10.06 |
| MA (Ours) | PerAct [34] | **84.00**±6.93 | 53.33±6.11 | 86.67±6.11 | **89.33**±6.11 | 8.00±4.00 | 33.33±2.31 |
| RLBench [33] | PerAct [34] | 58.67±50.80 | **53.33**±34.02 | **86.67**±12.86 | 84.00±13.86 | **30.67**±10.07 | **62.67**±9.24 |
| VoxPoser [3] | RVT-2 [60] | 5.33±6.11 | 1.33±2.31 | 1.33±2.31 | 2.67±2.31 | 1.33±2.31 | 17.33±2.31 |
| CAP [5] | RVT-2 [60] | 89.33±6.11 | - | - | 85.33±8.32 | 52.00±10.58 | 82.66±20.53 |
| Scaling-up [4] | RVT-2 [60] | 94.67±4.62 | 24.00±4.00 | 62.67±2.31 | 21.33±2.31 | 53.33±2.31 | 80.00±6.93 |
| MA (Ours) | RVT-2 [60] | 94.67±2.31 | **73.33**±2.31 | **93.33**±6.11 | 84.00±10.58 | 69.33±12.85 | **82.67**±12.22 |
| RLBench [33] | RVT-2 [60] | **97.33**±2.31 | 69.33±8.33 | 88.00±8.00 | **93.33**±4.62 | **72.00**±10.58 | 64.00±10.58 |

### 6.2.5 open jar

**Filename:** open_jar.py

**Task:** Uncap the green jar.

**Success Metric**: The green jar is out of its pre-defined capped location.

### 6.2.6 pickup cup

**Filename: Filename:** pickup_cup.py

**Task:** Pick up the red cup.

**Success Metric**: Lift up the red cup above the pre-defined location.

### 6.2.7 take umbrella

**Filename:** take_umbrella_out_of_stand.py

**Task:** Pick up the umbrella out of the umbrella stand.

**Success Metric**: Lift up the umbrella out of the umbrella stand.

### 6.2.8 sort mustard

**Filename:** sort_mustard.py

**Task:** Pick up the yellow mustard bottle, and place it into the red container.

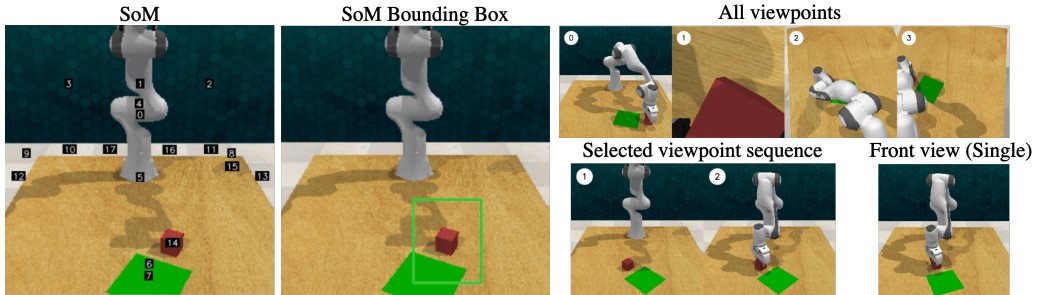

Figure 6: **Evaluation of visual prompting.** We systematic evaluate 5 different visual prompting techniques, and found that selected viewpoint sequence yields the highest performance.

**Success Metric**: The yellow mustard bottle inside red container.

### 6.2.9 `open wine`

**Filename:** `open_wine.py`

**Task:** Uncap the wine bottle.

**Success Metric**: The wine bottle cap is out of its original position.

### 6.2.10 `lamp on`

**Filename:** `lamp_on.py`

**Task:** Turn on the lamp.

**Success Metric**: The lamp light up.

### 6.2.11 `put knife`

**Filename:** `put_knife_on_chopping_board.py`

**Task:** Pick up the knife and place it onto the chopping board.

**Success Metric**: Knife on chopping board.

### 6.2.12 `push block`

**Filename:** `push_block_to_target.py`

**Task:** Push the red block down towards the green target.

**Success Metric**: The red block fails within the green target.

### 6.2.13 `insert block`

**Filename:** `insert_block.py`

**Task:** Push the green block into the jenga tower.

**Success Metric**: The green block inserted in.

### 6.2.14 `pick & lift`

**Filename:** `pick_and_lift.py`

**Task:** Pick up the red cube.

**Success Metric**: The red cube is lifted up.

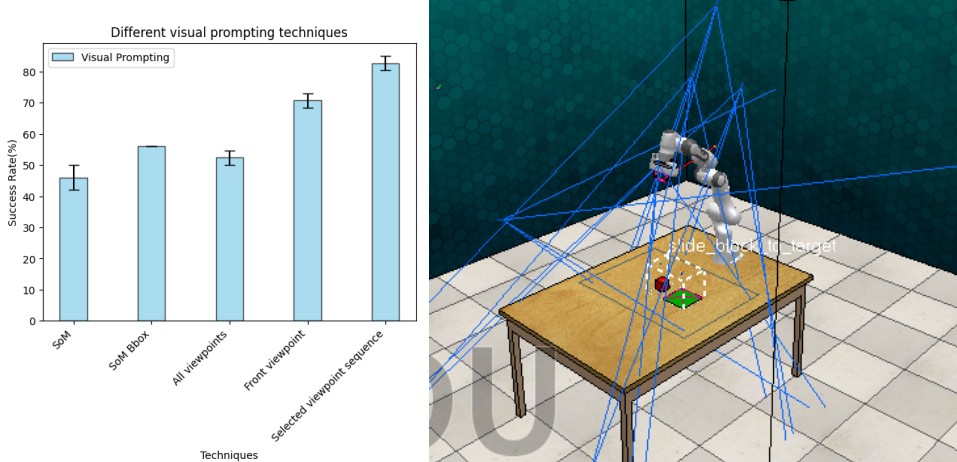

Figure 7: **Results for visual prompting techniques (Left).** We reported the various results for different visual prompting technique decision, and reported that selected viewpoint sequence yield the best performance. **Simulation scene setup (Right).** We leverage 4 different camera for evaluation.

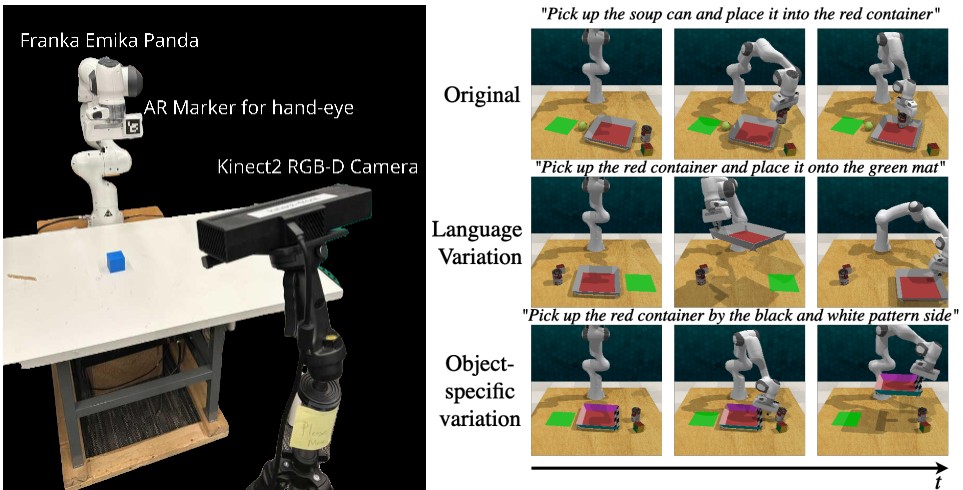

Figure 8: **Real-world experiment setup (Left).** We set up the real-world using this configuration. **Robustness and generalization evaluation.** We evaluated MANIPULATE-ANYTHING against VoxPoser for capability in generalizing to different language instructions and also object-specific manipulation.

## 6.3 Real-world experiments

### 6.3.1 Robot hardware setup

The real-robot experiments use a Franka Panda manipulator with a parallel gripper. For perception, we use a Kinect-2 RGB-D camera mounted on a tripod, at an angle, pointing towards the tabletop. Kinect-2 provides RGB-D images of resolution $512 \times 424$ at 30Hz. The extrinsic between the camera and robot base-frame are calibrated with the easy hand-eye package. We use an ARUCO AR marker mounted on the gripper to aid the calibration process, as shown in Figure 8.

### 6.3.2 Additional real-world everyday manipulation tasks

Beyond the five real-world experiments used for systematically evaluating MANIPULATE-ANYTHING, we also have additional real-world demonstrations generated in a zero-shot manner via MANIPULATE-ANYTHING. These demonstrations cover a range of tasks, from reasoning tasks to more precise everyday tasks. All of the tasks can be seen in Fig. 10. We also demonstrates that MANIPULATE-

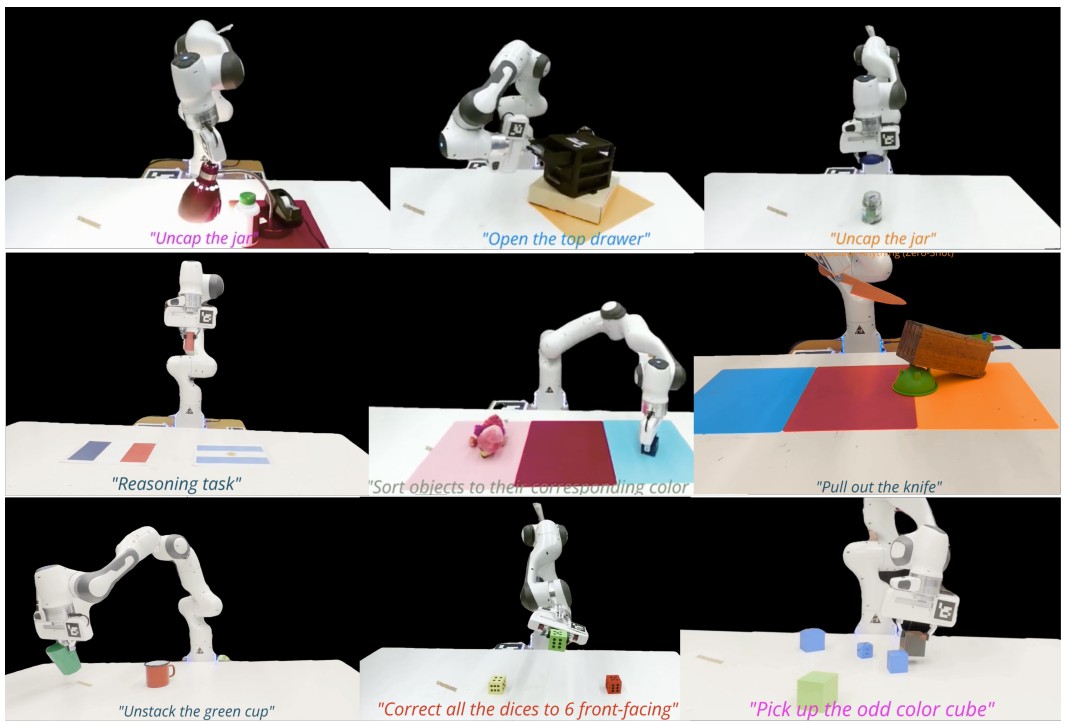

Figure 9: **More real-world experiments.**

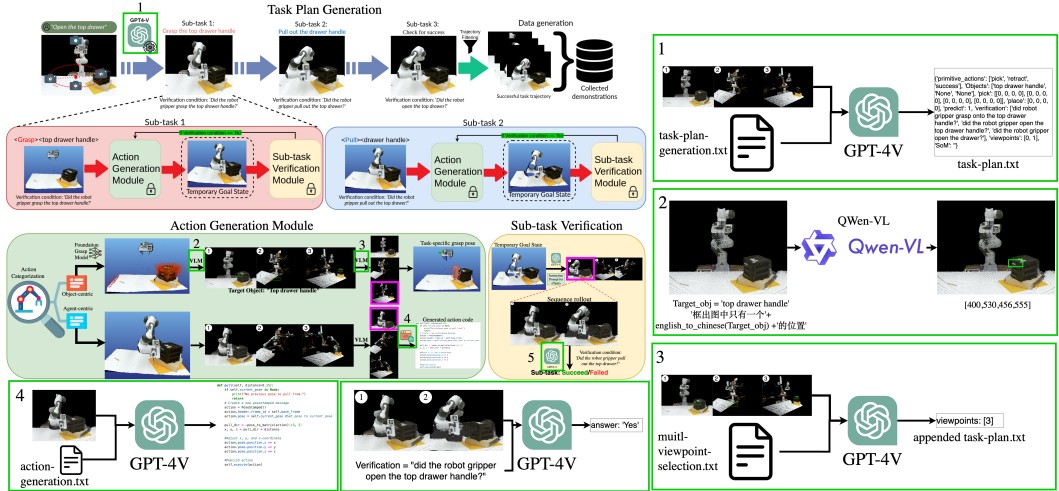

Figure 10: **In-depth Examination of VLMs for Manipulate-Anything.** The system consists of five components that utilize VLM API calls. We have provided a detailed breakdown of the inputs and outputs for each of these VLMs for reference.

ANYTHING can generate trajectories compares to hand-crafted trajectories by humans as depicted in Fig 11.

### 6.3.3 Baseline implementation details

For most of the baselines we followed the original implementation with minor modifications. For the `Code as Policies` baseline we re-implemented most of the environment code using `PyRep` instead of `PyBullet`. This includes the implementation of various motion primitives that form the

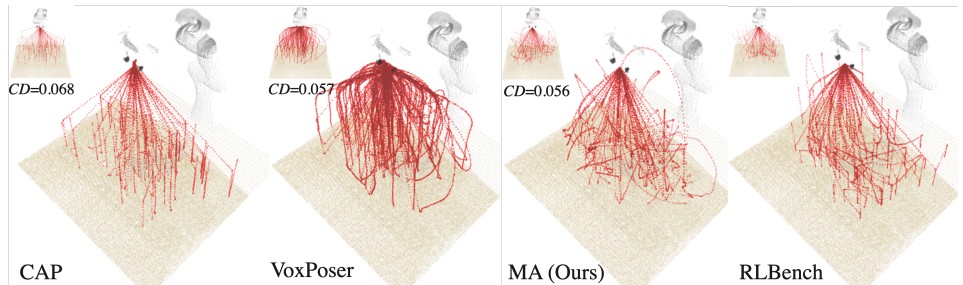

CD=0.068          CD=0.057          CD=0.056

CAP          VoxPoser          MA (Ours)          RLBench

Figure 11: **Action Distribution for Generated Data:** We compare the action distribution of data generated by various methods against human-generated demonstrations via RLBench on the same set of tasks. We observed a high similarity between the distribution of our generated data and the human-generated data. This is further supported by the computed CD between our methods and the RLBench data, which yields the lowest (CD=0.056).

exposed API to the language model program. For example, one of such primitives is shown in the following Figure 12.

## 6.4 Error Breakdown

We examine the potential errors introduced by each respective VLM and explore ways to further improve the overall system. We conducted simulation experiments where tasks could be easily reset, focusing on two major components where VLMs make decisions: "Perception error" and "Reasoning error." "Perception error" refers to mistakes made by the VLM in detecting target objects and selecting optimal viewpoints for generating task-specific grasp pose, primarily related to the selection of the Multi-viewpoint VLM. "Reasoning error" involves the Sub-task Verification Module, where the VLM is responsible for deciding if sub-tasks have been successfully completed. Due to resource constraints, we conducted experiments on the `play_jenga` task. We systematically replaced the VLMs in each component with a human, allowing the human to make decisions. By comparing the system's performance with human decision-making to that with VLMs, we were able to quantify the errors caused by the VLMs.

## 6.5 Additional Ablation Studies

We conducted two main set of ablation studies, we first look at how different visual prompting works for sub-goal verification, and then we further evaluated MANIPULATE-ANYTHING'S robustness and generalization to language instructions in another set of experiments.

For evaluating different visual prompts for sub-goal verification on the `put_block` task, we employed the following methods: 1) Set-of-Mark [62] on a single view, 2) Set-of-Mark with bounding box annotation, 3) Concatenated all viewpoints, 4) Front view only, and 5) Selected viewpoint sequence as shown in Fig. 6. We observed that the selected viewpoint sequences were the most effective in achieving correct sub-goal verification, obtaining the highest success rate as shown in Fig. 7.

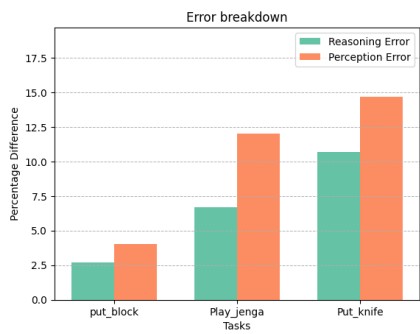

Figure 13: **Error breakdown.** Error breakdown for three tasks from simulation.

We further evaluated the generalization capabilities of our model in terms of object-specific manipulation and robustness to changes in language instructions. For language instruction variations, we altered the instructions for the same scene and found that MANIPULATE-ANYTHING outperforms VoxPoser by 60% over 25 episodes. For object-specific variations, where instructions targeted specific parts of objects, MANIPULATE-ANYTHING outperformed VoxPoser by 16%, as shown in Fig. 8.

### 6.6 Prompts

Prompts used for **multi-viewpoint VLM selection**, **task plan generation**, **action generation** and **sub-task verification** can be found below.

- **Multi-viewpoint VLM (Verification)**: Takes in 4 images concatenated into a single frame with number annotated on top and returns a selection number of the most optimal viewpoint for verfying the sub-task.
  Simulation: multi-viewpoint-VLM-selection.txt

- **Multi-viewpoint VLM (Object centric action):** Takes in 4 images concatenated into a single frame with number annotated on top and returns a selection number of the most optimal viewpoint for filter the grasp candidate poses. Simulation: Multi-viewpoint VLM-verification-object-centric.txt

- **Task plan generation:** Takes in a natural language instruction, and outputs a task plan in json file with the correct format.
  Simulation: task-plan-generation-prompt.txt

- **Sub-task verification:** Takes in the selected viewpoint rollout along with the verification condition, and outputs binary 'yes' or 'no'.
  Simulation: sub-task-verification-prompt.txt

- **Action-Generation:** Takes the action primitive and generates codes for executing the action in simulation.
  Simulation: action-generation.txt

### 6.7 Best Task Plans and Action Primitives

We ran MANIPULATE-ANYTHING with multi-processing during simulation to obtain the best set of task plans and action primitives for any given task. This set of information is then used to generate more data at scale for distilling a policy. We have compiled all the task plan JSON files and action primitive code that achieved the highest success rates in these tasks.
**Best Task Plans:** best-task-plans.txt
**Skills Library:** skills-library.txt

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

```python
    def primitive_pick_place(self, obj1_name: str, obj2_name: str) -> None:
        """Motion primitive used for picking and placing an object

        Parameters
        ----------
            obj1_name: str
                The name of the object to pick
            obj2_name: str
                The name of the object on which to place the first
        """
        obj1_name = obj1_name.replace(".", "_")
        obj2_name = obj2_name.replace(".", "_")

        pick_obj = self.get_obj_if_exists(obj1_name)
        assert pick_obj is not None
        place_obj = self.get_obj_if_exists(obj2_name)
        assert place_obj is not None

        pick_pos, pick_rot = self.get_grasp_pose(
            cast(Shape, pick_obj), self.get_ee_orientation()
        )
        place_pos, place_rot = self.get_place_pose(
            cast(Shape, place_obj), self.get_ee_orientation()
        )

        pre_pick_pos = post_grasp_pos = pick_pos + np.array([0.0, 0.0, 0.2])
        pre_place_pos = place_pos + np.array([0.0, 0.0, 0.2])

        # Move to pre-pick position
        self.movep(ee_xyz=pre_pick_pos, ee_euler=pick_rot)
        while not np.allclose(pre_pick_pos, self.get_ee_position(), atol=1e-2):
            self.step()

        # Close in to pick position
        self.movep(ee_xyz=pick_pos, ee_euler=pick_rot, ignore_collisions=True)
        while not np.allclose(pick_pos, self.get_ee_position(), atol=1e-2):
            self.step()

        # Close the gripper
        while not self._robot.gripper.actuate(0.0, 0.04):
            self.step()
        self._robot.gripper.grasp(pick_obj)

        # Move to post-grab position
        self.movep(post_grasp_pos, ignore_collisions=True)
        while not np.allclose(
            post_grasp_pos, self.get_ee_position(), atol=1e-2
        ):
            self.step()

        # Move to pre-place position (to avoid collisions)
        self.movep(
            ee_xyz=pre_place_pos, ee_euler=place_rot, ignore_collisions=True
        )
        while not np.allclose(pre_place_pos, self.get_ee_position(), atol=1e-2):
            self.step()

        # Move to place position
        self.movep(ee_xyz=place_pos, ee_euler=place_rot, ignore_collisions=True)
        while not np.allclose(place_pos, self.get_ee_position(), atol=1e-2):
            self.step()

        # Open the gripper to release the object
        self._robot.gripper.release()
        while not self._robot.gripper.actuate(1.0, 0.04):
            self.step()
```

Figure 12: **Example of one of the primitives implemented for Code as Policies**

