# OpenReview forum: "Manipulate-Anything: Automating Real-World Robots using Vision-Language Models"
_robot-learning.org/CoRL/2024/Conference — CoRL 2024_

### Official Review · Reviewer_3U2k · 2024-06-28
**Impressive results; writing needs improvement**

**Originality:** 3
**Technical Quality:** 4
**Clarity Of Presentation:** 3
**Potential Impact:** 3
**Recommendation:** 3
**Confidence:** 3

**Review:**

### Strengths
* The paper results are quite impressive. I’m especially impressed with the performance given that object models are not needed.
* The work is certainly relevant and is likely to be of considerable interest to the CoRL audience.
* The scope of the paper is ambitious. As discussed below, I think the writing needs to be improved, but the experiments are largely successful in achieving the ambitious scope.
* Among other clever choices, I like the idea to let the VLM decide which camera view to use. (I am not sure if this is an original idea, but I haven’t seen it before.)
* I also appreciated the ablation studies. There could be even more of them.

### Weaknesses
* My main concerns are related to clarity and reproducibility. Several important details are unclear; I will ask about them in the Questions section below. Also, no code was provided with the submission (but certain Python files are referenced in Appendix Section 2).
* A secondary concern is novelty. The paper ultimately joins together existing methods. But the experiments are sufficiently thorough and impressive to override this concern---it is a good systems paper.
* The name “Manipulate Anything” is likely to frustrate people in the manipulation community. The tasks considered in this paper are a small subset of manipulation. (I know that the name is an homage to Segment Anything and that other recent papers have followed this pattern.)
* One of the claimed advantages with respect to baselines is that MANIPULATE-ANYTHING doesn’t need “hand-designed skills”. I think this is overclaimed. The object-centric actions here are certainly hand-designed skills (that happen to be implemented with a grasp foundation model.) The agent-centric skills are arguably less hand-designed, but there is not enough information presented to assess whether the “program policies” here are meaningfully more sophisticated than the primitive actions. For example, does pulling out a shelf just involve one primitive action? Can that really be called a “skill”?
* The limitations section is extremely weak. The proposed approach is fairly involved; a number of design choices are made; and each choice has potential weaknesses. I know space is limited, but an extended limitations section could go in the appendix.
* The authors could have presented this work as just an open-world manipulation approach, but they instead describe it as that _and_ a data generation approach. I like this in theory, but I think the current execution is not entirely successful:
   * The justification is buried: “L188: Zero-shot methods like MANIPULATE-ANYTHING are computationally expensive but hold the potential to generate useful training data.” This should be said earlier.
   * Also, the computational expense of MANIPULATE-ANYTHING is not actually measured or discussed in detail.
   * As the paper says, one of the requirements for a data generation approach is that the data are diverse. But it is never really explained why MANIPULATE-ANYTHING would produce more diverse data than the baselines.
   * This more-involved narrative means that there is less room in the paper to clearly describe the main methods. It also means that some of the impressive aspects of the work are buried.
   * Readers of this paper may ultimately be confused about whether they should use MANIPULATE-ANYTHING as a general manipulation approach, or instead, as a data generator for some learning-based manipulation approach.
* Some of the videos on the supplemental website have issues:
   * One of the first videos is labeled “uncap the jar”, but the robot instead turns on a light. Is this a video labeling bug, or a failure example? It is presented as though it is meant to be a success.
   * The videos are not annotated with timing markers and appear to be highly edited.
   * The simulated version of “close box” seems to “jump” in such a way that avoids the robot needing to actually carefully release the box lid (which is an interesting aspect of the manipulation problem.)
   * The “pick up cup” version of PerAct (with MANIPULATE-ANYTHING) shows a physically unrealistic grasp. Other videos have similar issues.
   * Some of the videos appear to suddenly jitter to other scenes after the task is complete.
* The paper figures could be improved.
   * Figure 1 is not very informative and it takes up a lot of room. I didn’t get anything from it that wasn’t clear from the text. Also, when I first started looking at the paper, I did not realize that the “existing robot data generator” panel represents baselines; I thought it was part of the proposed approach.
   * Figure 2 is more informative but could be polished.
   * Figure 3 is largely redundant with the bottom half of Figure 2.
* I am glad that most of the experiments were run over multiple seeds, but 3 is insufficient.

### Minor
* L34: “abuz” → abuzz
* L38: extraneous period after [29]
* L47: “Thus it can easily generalized to the real world” grammar
* L48: “excute”
* Figure 1 caption: “priviledged”
* Check the rest of the paper carefully for typos and grammatical issues; there are many others.
* The paper says “zero-shot” so many times -- something like 17 times. L54 says “...  it can be prompted with a novel, never-before-seen task and complete it in a zero-shot manner.” Three different ways of saying the same thing!

**Quality Of The Limitations Section:**

2

**Questions For Rebuttal:**

1. In-context learning is used in multiple places. For each, what are the examples used for the context? Are these examples always the same, or are they task-dependent? How different from the test tasks are they?
2. The paper occasionally refers to a “target object” as part of the method. Is there an assumption that each natural language task description explicitly mentions one target object? If not, where do the target objects come from?
3. The appendix says that code snippets are “appended to a skill library for future use.” Is this talking about within a single task? What is the “future use” here?
4. L131 refers to a “detected bounding box” -- where in the pipeline is this bounding box detected?
5. How does the approach decide whether to use an object-centric or agent-centric action?
6. How many actions per subgoal are generated?
7. The paper mentions “recovery behavior into the collected demonstrations.” Does that mean that the entire collected trajectory is used for behavioral cloning? So the learned policy will also imitate the parts of the trajectory that lead to subgoal verification failures?
8. Which aspects of MANIPULATE-ANYTHING are responsible for generating diverse data?
9. Can you discuss and/or analyze the common failure modes of MANIPULATE-ANYTHING?

**Robotics Focus:**

4

**Summary Of Paper:**

This paper proposes a VLM-based method for manipulation and then uses the method to generate data for behavioral cloning.

**Summary Of Recommendation:**

Post rebuttal: the paper still has room for improvement in clarity, but I think it is above the bar for publication.

---

### Official Review · Reviewer_Ssao · 2024-07-20
**A framework utilizing VLMs and a grasp prediction model to address tasks within the limit of TAMP, while "an automated method for robot manipulation in real world environments" sounds like an overclaim**

**Originality:** 2
**Technical Quality:** 2
**Clarity Of Presentation:** 3
**Potential Impact:** 2
**Recommendation:** 2
**Confidence:** 3

**Review:**

## Strengths
1. The paper introduces using an off-the-shelf grasp prediction model to improve "action generation", which is commonly needed in similar frameworks relying on LLM/VLM for high-level or mid-level planning. The authors also introduce a sub-goal verification module to enhance "action generation".

## Weaknesses
1. It sounds like an overclaim that "Manipulate-Anything" is "a scalable automated generation method for real-world robotic manipulation". However, it can not manipulate "anything", even potentially. The existing frameworks that use LLMs or VLMs to generate plans or actions are mostly limited to which actions can be done. The ability to understand (detect or grasp) the concept of objects ("anything") is usually what LLMs or VLMs can provide, rather than the ability to "manipulate" objects. The tasks tackled in the paper are in general limited to quasi-static manipulation.
2. In my own opinion, the framework seems to be an LLM/VLM-based version of PDDLStream [1]. The error recovery only happens for trying different grasp poses, which is similar to sampling grasp poses in PDDLStream. The authors also agree that their method extends prior methods like TAMP in L79-L80. Thus, it is expected to see more quantitative comparison with TAMP methods. However, such comparison in experiments is missing. Although TAMP methods relies on privileged states in general, it seems not to be hard to replace privileged states with predicted states in the tasks evaluated in this paper. And there may exist prior works (see [2] for reference) that support online-planning (retry) or uncertainty.
3. The authors emphasize "real-world robotic manipulation" and "automated". However, main experiments are done in simulation and some important pain points in the real world are not addressed. For example, resetting environments is notoriously known as a bottleneck for real-world data collection, and how to recover from exploration or planning failure is not discussed in this paper. Besides, (correct me if wrong) RLBench does not fully simulate the dynamics (the grasp behavior is implemented as a constraint when the object is close enough to the gripper based on PyRep's proximity sensors). It makes the simulated experiments in the paper less useful for real-world applications.
4. The paper addresses some technical limitations of prior works, while some of them are hard to considered as a contribution. For example, VLMs might not be powerful enough when CAP and VoxPoser were done.
5. More related works can be included, like [3] and [4], for quantitative comparison.

- [1] Caelan R. Garrett, Tomás Lozano-Pérez, Leslie P. Kaelbling. PDDLStream: Integrating Symbolic Planners and Blackbox Samplers via Optimistic Adaptive Planning, International Conference on Automated Planning and Scheduling (ICAPS), 2020.
- [2] Garrett, Caelan Reed, et al. "Integrated task and motion planning." Annual review of control, robotics, and autonomous systems 4.1 (2021): 265-293.
- [3] Arenas, Montserrat Gonzalez, et al. "How to prompt your robot: A promptbook for manipulation skills with code as policies." Towards Generalist Robots: Learning Paradigms for Scalable Skill Acquisition@ CoRL2023. 2023.
- [4] Di Palo, Norman, and Edward Johns. "Keypoint Action tokens enable in-context imitation learning in robotics." arXiv preprint arXiv:2403.19578 (2024).

## Typos

There exist some typos in the paper.
1. L47: "Thus it can easily generalized to the real world."
2. Fig 4 caption: "12 tasks in simulation 5 tasks in the real world"
3. Fig 5 caption: "thehuman-generated"
4. L216: "tozero-shot"

**Quality Of The Limitations Section:**

1

**Questions For Rebuttal:**

1. It is unclear how the candidate grasp poses are filtered in L128-L129. Especially, how the grasp poses are represented for VLMs to understand (e.g., by text or visual)?
2. What few-shot demonstrations are used for in-context learning to select viewpoints (L23 in Appendix)? Are they the same for all tasks, or are task-specific demonstrations needed?
3. The authors mention several times "avoiding errors due to occlusion or ambiguity from a single viewpoint". However, if the image is re-rendered from the same point cloud (perhaps partial), it could not handle occlusion caused by the camera setup (e.g., 1 front camera in the real-world experiments). Choosing a different viewpoint may help VLMs, but can the authors use quantitative evidence to support the claim for avoiding errors due to occlusion?
4. For re-attempting the action generation step, what will happen if no actions lead to success? In L145-L146, it is said that "the next sub-goal is attempted", but it does not sound convincing if the skipped sub-goal is essential for the full task. And what will happen if an object is already grasped by the end-effector but with a "bad" grasp pose (e.g., the blade of a knife is held and thus the knife can not be inserted into the holder while VLMs happen to fail to find this issue when handling the previous sub-goal)?
5. Can the authors also use the ground truth simulation state information for zero-shot experiments in Sec 4.1? It can indicate how accurate the information of the target object’s asset names or positions is needed.
6. What are "human-generated demonstrations" in Sec 4.2? Are they tele-operated, or generated by hand-crafted scripts (e.g., provided by RLBench)? According to Table 2, it seems to be the latter.
7. Only PerAct is used for behavior cloning experiments while its limitations are also noticed by authors (e.g., poor for long-horizon memory). It is better to add more methods like ACT or Diffusion Policies to provide more comprehensive and convincing comparison.
8. Why is VoxPoser not included in Table 3? For 6 demonstrations generated by Manipulate-Anything (MA), do the authors repeatedly run MA until 6 demos are collected?
9. Which single task (L247) is used for the ablation study on scaling? Or is the success rate in Figure 6 averaged over 12 simulated tasks?
10. Can the authors provide error breakdown (like Sec 4.4 in VoxPoser) to analyze which aspects the proposed method improve over baselines?

**Robotics Focus:**

4

**Summary Of Paper:**

The paper proposes a framework, named "Manipulate-Anything", which leverages VLMs to plan a sequence of sub-goals (text) and an off-the-shelf grasp prediction model to generate grasp poses addressing sub-goals. A sub-goal verification module is introduced to support retry behaviors. It addresses some technical limitations in prior works like VoxPoser and Code as Policies.

**Summary Of Recommendation:**

The paper shows some technical improvements over prior works, while the framework and experiments fail to show that it can be scaled up.

---

### Official Review · Reviewer_1Hnu · 2024-07-23
**Well motivated, but mainly hampered by writing issues and fewer experiments than the claims imply.**

**Originality:** 3
**Technical Quality:** 2
**Clarity Of Presentation:** 2
**Potential Impact:** 3
**Recommendation:** 3
**Confidence:** 5

**Review:**

## Strengths

**Results:** Results are good, demonstrating significant gains over prior work in most of the tasks evaluated.

**Motivation:** This paper is well motivated, targeting *quantity, quality, and diversity* of training data. These are increasingly more important problems to tackle as models scale in size.

**Method:** The authors sensibly combined many existing frameworks together into one pipeline for data generation of robot manipulation data: grasping networks, VLMs, code generation, etc.

**Real world experiments:** It’s always great to see real world experiments at CoRL. These experiments are not vacuous and do demonstrate the ability of Manipulate Anything to actually work in real world tasks.

## Weaknesses

**Clarity Issues:**

- In the introduction, I don’t really get a sense of a “key idea” when reading. The 3rd paragraph contains many things that Manipulate-Anything enables but I am not sure what key insight drives the method to be better than the prior works mentioned in the 2nd paragraph. This isn’t a huge issue, but it would make the paper more clear.
- Figure 3: Everything is quite small yet there’s a lot of whitespace that could be filled.
- L124-131: Sentences in this paragraph seem a little disjoint, it could be re-written to flow more smoothly.
- L134: This multiple-viewpoint selection process seems like an important detail, perhaps a reference to a specific appendix section where this is discussed, or even better, a bit more detail here would be good (e.g., what does the prompt look like? how does the VLM choose?  etc.)
- Section 3.3: how does the multi-view VLM reasoning module work? What is the input/output? There is a lack of detail that doesn’t require that much extra space to fill and this seems too important to the method to be left to the appendix.
    - Following up on this, the appendix doesn’t actually contain that much detail for these method implementations. Example prompts/example responses are pretty standard in LLM for robotics papers, these should be included.
- Following up on the previous point, without the sense of a “key idea,” the method section is a little bit hard to understand as many low-level details are understandably left out due to space concerns. But, for example, if the key idea was to use VLMs to look at multiple viewpoints, then questions about minor implementation details that readers would have regarding other parts of the method would be less pressing. It would be good to rewrite just a little bit of the methods section overview and introduction to highlight a major idea which underpins the entire method and distinguishes it from all prior work, add extra details about this, and then according to space considerations other parts can be moved to the appendix as needed.

**Baselines:** VoxPoser and Code-as-Policies are both not really made for data generation. I think they’re valid comparisons, but likely not the best comparison for data generation for training policies. A better comparison would be https://arxiv.org/abs/2307.14535, which the authors already do cite in the related works but don’t really distinguish their method from too convincingly. I’m not specifically asking for this in the rebuttal period, but this does seem much better of a baseline.

**Experiments:** The claims made in the introduction about the *diversity* and *quantity* of possible data collected aren’t really supported by the somewhat limited set of experiments:

- There’s more than 12 tasks alone in RLBench, why just these 12 tasks?
- How about non-prehensile tasks?

**Overclaims:**

- There’s a claim of manipulating “any” static object but this isn’t really backed up by the experiments. I would remove this specific statement (abstract).
- In comparing to prior work: L32: Open-X contains *much* more than 20 tasks…. What is the definition of task here? If it’s the same as Manipulate Anything uses, then it should be far more than 20.

**Minor Issues:**

- L30 “human collection” → “human data collection”
- L34: “abuz” → “abuzz”
- L38: “.” → “,”
- L105: “We” → “we”
- Readability would be improved if appendix references also contained the actual sections they refer to.
- Some extra related work that seems quite relevant:
    - LLMs + offline RL for scaling manipulation data: https://clvrai.com/sprint
    - LLMs + motion planners for robotics to learn new tasks: https://mihdalal.github.io/planseqlearn/
    - Collecting demos with LLMs/VLMs: https://auto-rt.github.io/ — this paper is actually very relevant and should be contrasted against specifically in the related works.
    - LLMs +  RL for robotics to learn new tasks: https://clvrai.github.io/boss/
    - LLMs/VLMs for new task learning: https://rlvlmf2024.github.io/
    - Instruction augmentation with VLMs for data generation: https://instructionaugmentation.github.io/

**Quality Of The Limitations Section:**

2

**Questions For Rebuttal:**

Integrated in the review, please see the review.

**Robotics Focus:**

4

**Summary Of Paper:**

The authors propose Manipulate-Anything, a framework for generating data for robot manipulation which uses VLMs to help plan and execute tasks to generate data for policies to learn from.

**Summary Of Recommendation:**

I recommend a weak reject because the paper, while well-motivated and with some good results, suffers from writing issues and lacks comprehensive experimental validation to fully support its claims. Addressing these concerns would significantly strengthen the work and change my score.

---

### Author Rebuttal · Authors · 2024-08-10

We have attached a revised version of our submission and supplementary material (with changes in blue text) incorporating the reviewers' feedback.

Please see the Overall Response for a summary of the changes, and comments on the individual reviews for point-by-point responses to the reviewers' comments. We have also revised the videos and added additional experimental videos to the anonymous project page.

Thank you all!

---

### Decision · Program_Chairs · 2024-09-04

**Decision:**

Accept

**Comment:**

The paper presents a framework for manipulating objects by combining VLMs and foundation models to help plan and execute tasks.

Strengths: The reviewers agree on the importance of the problem setup. They also acknowledge the value of the real-world evaluation.

Weaknesses: The main criticism lies in the technical novelty of the method and its relationship to prior works, while concerns about overclaiming with the title. Improving the clarity and reproducability of the paper is suggested by  3U2k, Ssao and 1Hnu suggest stronger baseline methods among others.

Post response period: The authors have addressed most of reviewer's concerns. In particular, they have added baselines, more experiment videos and improved the overall clarity of the presentation. However, being able to manipulate any object still seems to be overclaiming. Given the author response, the updated ratings and their justifications, the AC recommends the paper to be accepted with a poster presentation.